# Decision Transformer: Reinforcement Learning via Sequence Modeling

**Lili Chen**[*,1], **Kevin Lu**[*,1], **Aravind Rajeswaran**[2], **Kimin Lee**[1],
**Aditya Grover**[2,3], **Michael Laskin**[1], **Pieter Abbeel**[1], **Aravind Srinivas**[†,4], **Igor Mordatch**[†,5]

[*]equal contribution   [†]equal advising

[1]UC Berkeley   [2]Facebook AI Research   [3]UCLA   [4]OpenAI   [5]Google Brain

`{lilichen, kzl}@berkeley.edu`

## Abstract

We introduce a framework that abstracts Reinforcement Learning (RL) as a sequence modeling problem. This allows us to draw upon the simplicity and scalability of the Transformer architecture, and associated advances in language modeling such as GPT-x and BERT. In particular, we present Decision Transformer, an architecture that casts the problem of RL as conditional sequence modeling. Unlike prior approaches to RL that fit value functions or compute policy gradients, Decision Transformer simply outputs the optimal actions by leveraging a causally masked Transformer. By conditioning an autoregressive model on the desired return (reward), past states, and actions, our Decision Transformer model can generate future actions that achieve the desired return. Despite its simplicity, Decision Transformer matches or exceeds the performance of state-of-the-art model-free offline RL baselines on Atari, OpenAI Gym, and Key-to-Door tasks.

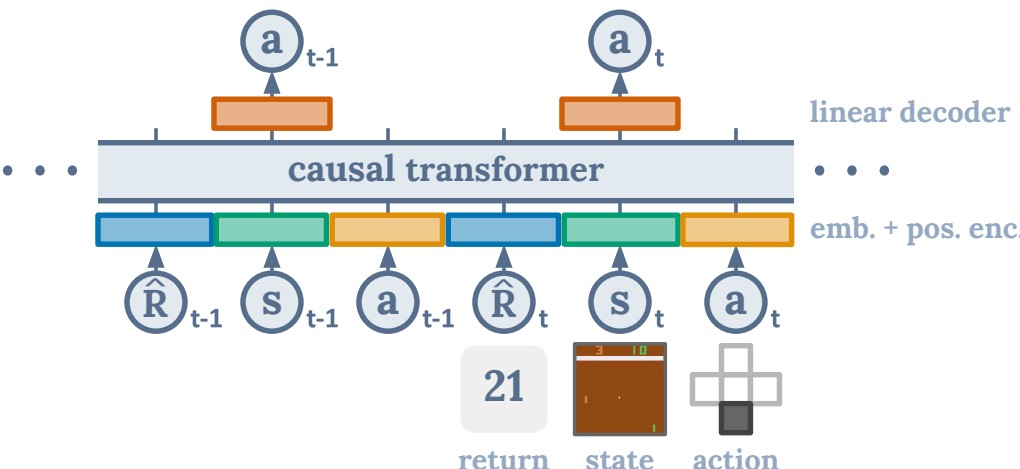

Figure 1: Decision Transformer architecture[1]. States, actions, and returns are fed into modality-specific linear embeddings and a positional episodic timestep encoding is added. Tokens are fed into a GPT architecture which predicts actions autoregressively using a causal self-attention mask.

---

[1]Our code is available at: `https://sites.google.com/berkeley.edu/decision-transformer`

35th Conference on Neural Information Processing Systems (NeurIPS 2021).

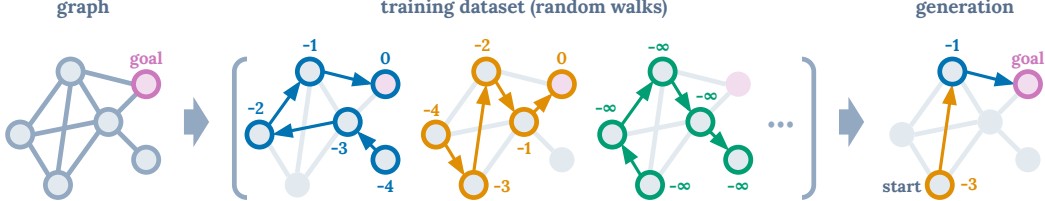

Figure 2: Illustrative example of finding shortest path for a fixed graph (left) posed as reinforcement learning. Training dataset consists of random walk trajectories and their per-node returns-to-go (middle). Conditioned on a starting state and generating largest possible return at each node, Decision Transformer sequences optimal paths.

# 1   Introduction

Recent work has shown transformers [1] can model large-scale distributions of semantic concepts, including capable zero-shot generalization in language [2] and impressive out-of-distribution image generation [3]. This stands in sharp contrast to much work in reinforcement learning (RL), which learns a single policy to model a particular narrow behavior distribution. Given the diversity of applications and impact of transformer models, we seek to examine their application to sequential decision making problems. In particular, instead of using transformers as an architectural choice for traditional RL algorithms [4, 5], we seek to study if trajectory modeling (analogous to language modeling) can serve as a *replacement* for conventional RL algorithms.

We consider the following shift in paradigm: instead of training a policy through conventional RL algorithms like temporal difference (TD) learning [6], the dominant paradigm in RL, we will train transformer models on collected experience using a sequence modeling objective. This will allow us to bypass the need for bootstrapping to propagate returns – thereby avoiding one of the "deadly triad" [6] known to destabilize RL. It also avoids the need for discounting future rewards, as typically done in TD-learning, which can induce undesirable short-sighted behaviors. Additionally, we can make use of existing transformer frameworks widely used in language and vision that are easy to scale, utilizing a large body of work studying stable training of transformer models; this approach removes the need for specialized RL frameworks by appealing only to commonplace supervised learning systems. Given their demonstrated ability to model long sequences and wide data distributions, transformers also have other advantages. Transformers can perform credit assignment directly via self-attention, in contrast to Bellman backups which slowly propagate rewards and are prone to "distractor" signals [7]. This can enable transformers to still work effectively in the presence of sparse or distracting rewards. Furthermore, a transformer modeling approach can model a wide distribution of behaviors, enabling better generalization and transfer. While "upside-down" reinforcement learning (UDRL) [8, 9, 10] also uses a supervised loss conditioned on a target return, our work is motivated by sequence modeling rather than supervised learning and seeks to benefit from modeling long sequences of behaviors. See Section 6 for more discussions about related works.

We explore our hypothesis by considering offline RL, where we will task agents with learning policies from suboptimal data – producing maximally effective behavior from fixed, limited experience. This task is traditionally challenging due to error propagation and value overestimation [11]. However, it is a natural task when training with a sequence modeling objective. By training an autoregressive model on sequences of states, actions, and returns, we reduce policy sampling to autoregressive generative modeling. We can specify the expertise of the policy – which "skill" to query – by manually setting the return tokens, acting as a prompt for generation.

**Illustrative example.** To get an intuition for our proposal, consider the task of finding a shortest path on a directed graph posed as an RL problem. The reward is $0$ when at the goal node and $-1$ otherwise. We train a GPT [12] model to predict next token in a sequence of returns-to-go (sum of future rewards), states, and actions. Training only on random walk data – with no expert demonstrations – we can at test time generate *optimal* trajectories by adding a prior to generate highest possible returns (see more details and empirical results in the Appendix) and subsequently generate actions

conditioned on that. Thus, by combining the tools of sequence modeling with hindsight return information, we achieve policy improvement without the need for dynamic programming.

Motivated by this observation, we propose Decision Transformer, where we use the GPT architecture to autoregressively model trajectories (shown in Figure 1). We study whether sequence modeling can perform policy optimization by evaluating Decision Transformer on offline RL benchmarks in Atari [13], OpenAI Gym [14], and Key-to-Door [15] environments. We show that – *without using dynamic programming* – Decision Transformer performs comparably on these benchmarks to state-of-the-art model-free offline RL algorithms [16, 17]. Furthermore, in tasks where long-term credit assignment is required, Decision Transformer capably outperforms RL algorithms. With this work, we hope to bridge vast recent progress in transformer models with RL problems.

## 2 Preliminaries

### 2.1 Offline reinforcement learning

We consider learning in a Markov decision process (MDP) described by the tuple $(\mathcal{S}, \mathcal{A}, P, \mathcal{R})$. The MDP tuple consists of states $s \in \mathcal{S}$, actions $a \in \mathcal{A}$, transition dynamics $P(s'|s, a)$, and a reward function $r = \mathcal{R}(s, a)$. We use $s_t$, $a_t$, and $r_t = \mathcal{R}(s_t, a_t)$ to denote the state, action, and reward at timestep $t$, respectively. The goal in reinforcement learning is to learn a policy which maximizes the expected return $\mathbb{E}\left[\sum_{t=1}^{T} r_t\right]$ in an MDP. In offline reinforcement learning, instead of obtaining data via environment interactions, we only have access to some fixed limited dataset consisting of trajectories from the environment. This setting is harder as it removes the ability for agents to explore the environment and collect additional feedback.

### 2.2 Transformers

Transformers were proposed by Vaswani et al. [1] as an architecture to efficiently model sequences. They consist of stacked self-attention layers with residual connections. Each self-attention layer receives $n$ embeddings $\{x_i\}_{i=1}^{n}$ corresponding to unique input tokens, and outputs $n$ embeddings $\{z_i\}_{i=1}^{n}$, preserving the input dimensions. The $i$-th token is mapped via linear transformations to a key $k_i$, query $q_i$, and value $v_i$. The $i$-th output of the self-attention layer is given by weighting the values $v_j$ by the normalized dot product between the query $q_i$ and other keys $k_j$:

$$z_i = \sum_{j=1}^{n} \texttt{softmax}(\{\langle q_i, k_{j'} \rangle\}_{j'=1}^{n})_j \cdot v_j. \tag{1}$$

This allows the layer to assign "credit" by implicitly forming state-return associations via similarity of the query and key vectors (maximizing the dot product). In this work, we use the GPT architecture [12], which modifies the transformer architecture with a causal self-attention mask to enable autoregressive generation, replacing the summation/softmax over the $n$ tokens with only the previous tokens in the sequence ($j \in [1, i]$). We defer the other architecture details to the original papers.

## 3 Method

In this section, we present Decision Transformer, which models trajectories autoregressively with minimal modification to the transformer architecture, as summarized in Figure 1 and Algorithm 1.

**Trajectory representation.** The key desiderata in our choice of trajectory representation are (a) it should enable transformers to learn meaningful patterns and (b) we should be able to conditionally generate actions at test time. It is nontrivial to model rewards since we would like the model to generate actions based on *future* desired returns, rather than past rewards. As a result, instead of modeling the rewards directly, we model the returns-to-go $\widehat{R}_t = \sum_{t'=t}^{T} r_{t'}$. This leads to the following trajectory representation which is amenable to autoregressive training and generation:

$$\tau = \left( \widehat{R}_1, s_1, a_1, \widehat{R}_2, s_2, a_2, \ldots, \widehat{R}_T, s_T, a_T \right). \tag{2}$$

**Architecture.** We feed the last $K$ timesteps into Decision Transformer, for a total of $3K$ tokens (one for each modality: return-to-go, state, or action). To obtain token embeddings, we learn a linear

layer for each modality, which projects raw inputs to the embedding dimension, followed by layer normalization [18]. For environments with visual inputs, the state is fed into a convolutional encoder instead of a linear layer. Additionally, an embedding for each timestep is learned and added to each token – note this is different than the standard positional embedding used by transformers, as one timestep corresponds to three tokens. The tokens are then processed by a GPT [12] model, which predicts future action tokens via autoregressive modeling.

**Training.** We sample minibatches of sequence length $K$ from the dataset. The prediction head corresponding to the input token $s_t$ is trained to predict $a_t$ – either with cross-entropy loss for discrete actions or mean-squared error for continuous actions – and the losses for each timestep are averaged. We did not find predicting the states or returns-to-go to be necessary for good performance, although it is possible (as shown in Section 5.3) and would be an interesting study for future work.

**Evaluation.** During evaluation rollouts, we specify a target return based on our desired performance (e.g., specify maximum possible return to generate expert behavior) as well as the environment starting state, to initialize generation. After executing the generated action, we decrement the target return by the achieved reward and obtain the next state. We repeat this process of generating actions and applying them to obtain the next return-to-go and state until episode termination.

---

**Algorithm 1** Decision Transformer Pseudocode (for continuous actions)

```
# R, s, a, t: returns-to-go, states, actions, or timesteps
# K: context length (length of each input to DecisionTransformer)
# transformer: transformer with causal masking (GPT)
# embed_s, embed_a, embed_R: linear embedding layers
# embed_t: learned episode positional embedding
# pred_a: linear action prediction layer

# main model
def DecisionTransformer(R, s, a, t):
    # compute embeddings for tokens
    pos_embedding = embed_t(t)  # per-timestep (note: not per-token)
    s_embedding = embed_s(s) + pos_embedding
    a_embedding = embed_a(a) + pos_embedding
    R_embedding = embed_R(R) + pos_embedding

    # interleave tokens as (R_1, s_1, a_1, ..., R_K, s_K)
    input_embeds = stack(R_embedding, s_embedding, a_embedding)

    # use transformer to get hidden states
    hidden_states = transformer(input_embeds=input_embeds)

    # select hidden states for action prediction tokens
    a_hidden = unstack(hidden_states).actions

    # predict action
    return pred_a(a_hidden)

# training loop
for (R, s, a, t) in dataloader:  # dims: (batch_size, K, dim)
    a_preds = DecisionTransformer(R, s, a, t)
    loss = mean((a_preds - a)**2)  # L2 loss for continuous actions
    optimizer.zero_grad(); loss.backward(); optimizer.step()

# evaluation loop
target_return = 1  # for instance, expert-level return
R, s, a, t, done = [target_return], [env.reset()], [], [1], False
while not done:  # autoregressive generation/sampling
    # sample next action
    action = DecisionTransformer(R, s, a, t)[-1]  # for cts actions
    new_s, r, done, _ = env.step(action)

    # append new tokens to sequence
    R = R + [R[-1] - r]  # decrement returns-to-go with reward
    s, a, t = s + [new_s], a + [action], t + [len(R)]
    R, s, a, t = R[-K:], ...  # only keep context length of K
```

---

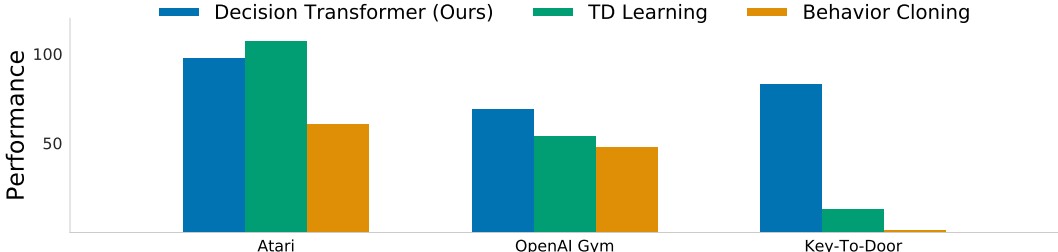

Figure 3: Results comparing Decision Transformer (ours) to TD learning (CQL) and behavior cloning across Atari, OpenAI Gym, and Minigrid. On a diverse set of tasks, Decision Transformer performs comparably or better than traditional approaches.

## 4 Evaluations on offline RL benchmarks

In this section, we investigate if Decision Transformer can perform well compared to standard TD and imitation learning approaches for offline RL. TD learning algorithms represent the conventional state-of-the-art, while imitation learning algorithms have similar formulations to Decision Transformer. The exact algorithms depend on the environment but our motivations are as follows:

- **TD learning**: most of these methods use an action-space constraint or value pessimism, and will be the most faithful comparison to Decision Transformer, representing standard RL methods. A state-of-the-art model-free method is Conservative Q-Learning (CQL) [17] which serves as our primary comparison. In addition, we also compare against other prior model-free RL algorithms like BEAR [19] and BRAC [20].

- **Imitation learning**: this regime similarly uses supervised losses for training, rather than Bellman backups. We use behavior cloning here, and include a more detailed discussion in Section 5.1.

We evaluate on both discrete (Atari [13]) and continuous (OpenAI Gym [14]) control tasks. The former requires long-term credit assignment, while the latter requires fine-grained continuous control, representing a diverse set of tasks. Our main results are summarized in Figure 3, where we show averaged expert normalized performance for each domain.

### 4.1 Atari

The Atari benchmark is challenging due to its high-dimensional visual inputs and difficulty of credit assignment arising from the delay between actions and resulting rewards. We evaluate our method on 1% of all samples in the DQN-replay dataset as per Agarwal et al. [16], representing 500 thousand of the 50 million transitions observed by an online DQN agent [21] during training; we report the mean and standard deviation of 3 seeds. We normalize scores based on a professional gamer, following the protocol of Hafner et al. [22], where 100 represents the professional gamer score and 0 represents a random policy.

We compare to CQL [17], REM [16], and QR-DQN [23] on four Atari tasks (Breakout, Qbert, Pong, and Seaquest) that are evaluated in Agarwal et al. [16]. We use context lengths of $K = 30$ for Decision Transformer (except $K = 50$ for Pong); for results with different values of $K$ see the supplementary material. We also report the performance of behavior cloning (BC), which utilizes

| Game | DT (Ours) | CQL | QR-DQN | REM | BC |
|------|-----------|-----|--------|-----|-----|
| Breakout | $\mathbf{267.5 \pm 97.5}$ | 211.1 | 21.1 | 32.1 | $138.9 \pm 61.7$ |
| Qbert | $15.1 \pm 11.4$ | $\mathbf{104.2}$ | 1.7 | 1.4 | $17.3 \pm 14.7$ |
| Pong | $106.1 \pm 8.1$ | $\mathbf{111.9}$ | 20.0 | 39.1 | $85.2 \pm 20.0$ |
| Seaquest | $\mathbf{2.4 \pm 0.7}$ | 1.7 | 1.4 | 1.0 | $2.1 \pm 0.3$ |

Table 1: Gamer-normalized scores for the 1% DQN-replay Atari dataset. We report the mean and variance across 3 seeds. Best mean scores are highlighted in bold. Decision Transformer (DT) performs comparably to CQL on 3 out of 4 games, and outperforms other baselines in most games.

the same network architecture and hyperparameters as Decision Transformer but does not have return-to-go conditioning[2]. For CQL, REM, and QR-DQN baselines, we report numbers directly from the CQL paper. We show results in Table 1. Our method is competitive with CQL in 3 out of 4 games and outperforms or matches REM, QR-DQN, and BC on all 4 games.

## 4.2 OpenAI Gym

In this section, we consider the continuous control tasks from the D4RL benchmark [24]. We also consider a 2D reacher environment that is not part of the benchmark, and generate the datasets using a similar methodology to the D4RL benchmark. Reacher is a goal-conditioned task and has sparse rewards, so it represents a different setting than the standard locomotion environments (HalfCheetah, Hopper, and Walker). The different dataset settings are described below.

1. Medium: 1 million timesteps generated by a "medium" policy that achieves approximately one-third the score of an expert policy.

2. Medium-Replay: the replay buffer of an agent trained to the performance of a medium policy (approximately 25k-400k timesteps in our environments).

3. Medium-Expert: 1 million timesteps generated by the medium policy concatenated with 1 million timesteps generated by an expert policy.

We compare to CQL [17], BEAR [19], BRAC [20], and AWR [25]. CQL represents the state-of-the-art in model-free offline RL, an instantiation of TD learning with value pessimism. Score are normalized so that 100 represents an expert policy, as per Fu et al. [24]. CQL numbers are reported from the original paper; BC numbers are run by us; and the other methods are reported from the D4RL paper. Our results are shown in Table 2. Decision Transformer achieves the highest scores in a majority of the tasks and is competitive with the state of the art in the remaining tasks.

| Dataset | Environment | DT (Ours) | CQL | BEAR | BRAC-v | AWR | BC |
|---------|-------------|-----------|-----|------|--------|-----|-----|
| Medium-Expert | HalfCheetah | $\mathbf{86.8 \pm 1.3}$ | 62.4 | 53.4 | 41.9 | 52.7 | 59.9 |
| Medium-Expert | Hopper | $107.6 \pm 1.8$ | $\mathbf{111.0}$ | 96.3 | 0.8 | 27.1 | 79.6 |
| Medium-Expert | Walker | $\mathbf{108.1 \pm 0.2}$ | 98.7 | 40.1 | 81.6 | 53.8 | 36.6 |
| Medium-Expert | Reacher | $\mathbf{89.1 \pm 1.3}$ | 30.6 | - | - | - | 73.3 |
| Medium | HalfCheetah | $42.6 \pm 0.1$ | 44.4 | 41.7 | $\mathbf{46.3}$ | 37.4 | 43.1 |
| Medium | Hopper | $\mathbf{67.6 \pm 1.0}$ | 58.0 | 52.1 | 31.1 | 35.9 | 63.9 |
| Medium | Walker | $74.0 \pm 1.4$ | 79.2 | 59.1 | $\mathbf{81.1}$ | 17.4 | 77.3 |
| Medium | Reacher | $51.2 \pm 3.4$ | 26.0 | - | - | - | $\mathbf{48.9}$ |
| Medium-Replay | HalfCheetah | $36.6 \pm 0.8$ | 46.2 | 38.6 | $\mathbf{47.7}$ | 40.3 | 4.3 |
| Medium-Replay | Hopper | $\mathbf{82.7 \pm 7.0}$ | 48.6 | 33.7 | 0.6 | 28.4 | 27.6 |
| Medium-Replay | Walker | $\mathbf{66.6 \pm 3.0}$ | 26.7 | 19.2 | 0.9 | 15.5 | 36.9 |
| Medium-Replay | Reacher | $18.0 \pm 2.4$ | $\mathbf{19.0}$ | - | - | - | 5.4 |
| **Average (Without Reacher)** | | **74.7** | 63.9 | 48.2 | 36.9 | 34.3 | 46.4 |
| **Average (All Settings)** | | **69.2** | 54.2 | - | - | - | 47.7 |

Table 2: Results for D4RL datasets[4]. We report the mean and variance for three seeds. Decision Transformer (DT) outperforms conventional RL algorithms on almost all tasks.

## 5 Discussion

### 5.1 Does Decision Transformer perform behavior cloning on a subset of the data?

In this section, we seek to gain insight into whether Decision Transformer can be thought of as performing imitation learning on a subset of the data with a certain return. To investigate this, we propose a new method, Percentile Behavior Cloning (%BC), where we run behavior cloning on only the top $X\%$ of timesteps in the dataset, ordered by episode returns. The percentile $X\%$ interpolates between standard BC ($X = 100\%$) that trains on the entire dataset and only cloning the best observed

---

[2]We also tried using an MLP with $K = 1$ as in prior work, but found this was worse than the transformer.

[4]Given that CQL is generally the strongest TD learning method, for Reacher we only run the CQL baseline.

| Dataset | Environment | DT (Ours) | 10%BC | 25%BC | 40%BC | 100%BC | CQL |
|---------|-------------|-----------|-------|-------|-------|--------|-----|
| Medium | HalfCheetah | $42.6 \pm 0.1$ | 42.9 | 43.0 | 43.1 | 43.1 | **44.4** |
| Medium | Hopper | **$67.6 \pm 1.0$** | 65.9 | 65.2 | 65.3 | 63.9 | 58.0 |
| Medium | Walker | $74.0 \pm 1.4$ | 78.8 | **80.9** | 78.8 | 77.3 | 79.2 |
| Medium | Reacher | $51.2 \pm 3.4$ | 51.0 | 48.9 | 58.2 | **58.4** | 26.0 |
| Medium-Replay | HalfCheetah | $36.6 \pm 0.8$ | 40.8 | 40.9 | 41.1 | 4.3 | **46.2** |
| Medium-Replay | Hopper | **$82.7 \pm 7.0$** | 70.6 | 58.6 | 31.0 | 27.6 | 48.6 |
| Medium-Replay | Walker | $66.6 \pm 3.0$ | **70.4** | 67.8 | 67.2 | 36.9 | 26.7 |
| Medium-Replay | Reacher | $18.0 \pm 2.4$ | **33.1** | 16.2 | 10.7 | 5.4 | 19.0 |
| **Average** | | 56.1 | **56.7** | 52.7 | 49.4 | 39.5 | 43.5 |

Table 3: Comparison between Decision Transformer (DT) and Percentile Behavior Cloning (%BC).

trajectory ($X \to 0\%$), trading off between better generalization by training on more data with training a specialized model that focuses on a desirable subset of the data.

We show full results comparing %BC to Decision Transformer and CQL in Table 3, sweeping over $X \in [10\%, 25\%, 40\%, 100\%]$. Note that while both %BC and DT introduce hyperparameters, returns are human interpretable and it is relatively natural for humans to specify a desired return compared to choosing an optimal subset for cloning. When data is plentiful – as in the D4RL regime – we find %BC can match or beat other offline RL methods. On most environments, Decision Transformer is competitive with the performance of the best %BC, indicating it can hone in on a particular subset after training on the entire dataset distribution.

In contrast, when we study low data regimes – such as Atari, where we use 1% of a replay buffer as the dataset – %BC is weak (shown in Table 4). This suggests that in scenarios with relatively low amounts of data, Decision Transformer can outperform %BC by using all trajectories in the dataset to improve generalization, even if those trajectories are dissimilar from the return conditioning target. Our results indicate that Decision Transformer can be more effective than simply performing imitation learning on a subset of the dataset. On the tasks we considered, Decision Transformer either outperforms or is competitive to %BC, without the confound of having to select the optimal subset.

| Game | DT (Ours) | 10%BC | 25%BC | 40%BC | 100%BC |
|------|-----------|-------|-------|-------|--------|
| Breakout | **$267.5 \pm 97.5$** | $28.5 \pm 8.2$ | $73.5 \pm 6.4$ | $108.2 \pm 67.5$ | $138.9 \pm 61.7$ |
| Qbert | $15.1 \pm 11.4$ | $6.6 \pm 1.7$ | $16.0 \pm 13.8$ | $11.8 \pm 5.8$ | **$17.3 \pm 14.7$** |
| Pong | **$106.1 \pm 8.1$** | $2.5 \pm 0.2$ | $13.3 \pm 2.7$ | $72.7 \pm 13.3$ | $85.2 \pm 20.0$ |
| Seaquest | **$2.4 \pm 0.7$** | $1.1 \pm 0.2$ | $1.1 \pm 0.2$ | $1.6 \pm 0.4$ | $2.1 \pm 0.3$ |

Table 4: %BC scores for Atari. We report the mean and variance across 3 seeds. Decision Transformer (DT) outperforms all versions of %BC in most games.

## 5.2 How well does Decision Transformer model the distribution of returns?

We evaluate the ability of Decision Transformer to understand return-to-go tokens by varying the desired target return over a wide range – evaluating the multi-task distribution modeling capability of transformers. Figure 4 shows the average sampled return accumulated by the agent over the course of the evaluation episode for varying values of target return. On every task, the desired target returns and the true observed returns are highly correlated. On some tasks like Pong, HalfCheetah and Walker, Decision Transformer generates trajectories that almost perfectly match the desired returns (as indicated by the overlap with the oracle line). Furthermore, on some Atari tasks like Seaquest, we can prompt the Decision Transformer with higher returns than the maximum episode return available in the dataset, demonstrating that Decision Transformer is sometimes capable of extrapolation.

## 5.3 Does Decision Transformer perform effective long-term credit assignment?

To evaluate long-term credit assignment capabilities of our model, we consider a variant of the Key-to-Door environment proposed in Mesnard et al. [15]. This is a grid-based environment with a sequence of three phases: (1) in the first phase, the agent is placed in a room with a key; (2) then, the

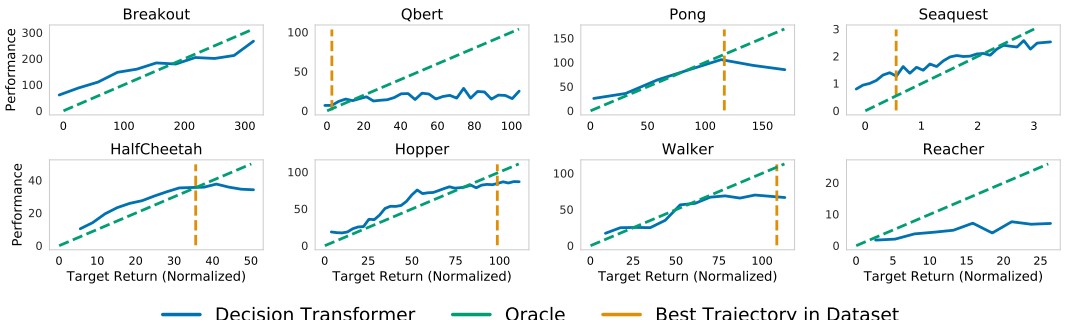

Figure 4: Sampled (evaluation) returns accumulated by Decision Transformer when conditioned on the specified target (desired) returns. **Top:** Atari. **Bottom:** D4RL medium-replay datasets.

agent is placed in an empty room; (3) and finally, the agent is placed in a room with a door. The agent receives a binary reward when reaching the door in the third phase, but **only** if it picked up the key in the first phase. This problem is difficult for credit assignment because credit must be propagated from the beginning to the end of the episode, skipping over actions taken in the middle.

We train on datasets of trajectories generated by applying random actions and report success rates in Table 5. Furthermore, for the Key-to-Door environment we use the entire episode length as the context, rather than having a fixed content window as in the other environments. Methods that use highsight return information: our Decision Transformer model and %BC (trained only on successful episodes) are able to learn effective policies – producing near-optimal paths, despite only training on random walks. TD learning (CQL) cannot effectively propagate Q-values over the long horizons involved and gets poor performance.

| Dataset | DT (Ours) | CQL | BC | %BC | Random |
|---|---|---|---|---|---|
| 1K Random Trajectories | **71.8**% | 13.1% | 1.4% | 69.9% | 3.1% |
| 10K Random Trajectories | 94.6% | 13.3% | 1.6% | **95.1**% | 3.1% |

Table 5: Success rate for Key-to-Door environment. Methods using hindsight (Decision Transformer, %BC) can learn successful policies, while TD learning struggles to perform credit assignment.

## 5.4 Can transformers be accurate critics in sparse reward settings?

In previous sections, we established that decision transformer can produce effective policies (actors). We now evaluate whether transformer models can also be effective critics. We modify Decision Transformer to output return tokens in addition to action tokens on the Key-to-Door environment. We find that the transformer continuously updates reward probability based on events during the episode, shown in Figure 5 (Left). Furthermore, we find the transformer attends to critical events in the episode (picking up the key or reaching the door), shown in Figure 5 (Right), indicating formation of state-reward associations as discussed in Raposo et al. [26] and enabling accurate value prediction.

## 5.5 Does Decision Transformer perform well in sparse reward settings?

A known weakness of TD learning algorithms is that they require densely populated rewards in order to perform well, which can be unrealistic and/or expensive. In contrast, Decision Transformer can improve robustness in these settings since it makes minimal assumptions on the density of the reward. To evaluate this, we consider a delayed return version of the D4RL benchmarks where the agent does not receive any rewards along the trajectory, and instead receives the cumulative reward of the trajectory in the final timestep. Our results for delayed returns are shown in Table 6. Delayed returns minimally affect Decision Transformer; and due to the nature of the training process, while imitation learning methods are reward agnostic. While TD learning collapses, Decision Transformer and %BC still perform well, indicating that Decision Transformer can be more robust to delayed rewards.

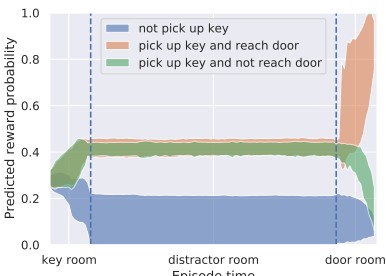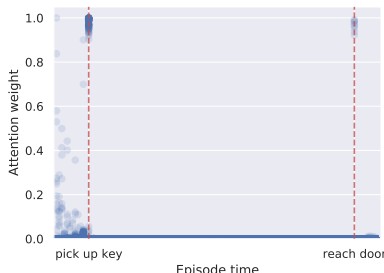

Figure 5: **Left:** Averages of running return probabilities predicted by the transformer model for three types of episode outcomes. **Right:** Transformer attention weights from all timesteps superimposed for a particular successful episode. The model attends to steps near pivotal events in the episode, such as picking up the key and reaching the door.

| Dataset | Environment | Delayed (Sparse) | | Agnostic | | Original (Dense) | |
|---|---|---|---|---|---|---|---|
| | | DT (Ours) | CQL | BC | %BC | DT (Ours) | CQL |
| Medium-Expert | Hopper | **107.3 ± 3.5** | 9.0 | 59.9 | 102.6 | 107.6 | 111.0 |
| Medium | Hopper | 60.7 ± 4.5 | 5.2 | 63.9 | **65.9** | 67.6 | 58.0 |
| Medium-Replay | Hopper | **78.5 ± 3.7** | 2.0 | 27.6 | 70.6 | 82.7 | 48.6 |

Table 6: Results for D4RL datasets with delayed (sparse) reward. Decision Transformer (DT) and imitation learning are minimally affected by the removal of dense rewards, while CQL fails.

## 5.6 Additional Discussions

For more discussions see the supplementary material.

## 6 Related work

**Offline reinforcement learning.** To mitigate the impact of distribution shift in offline RL, prior algorithms either (a) constrain the policy action space [27, 28, 29] or (b) incorporate value pessimism [27, 17], or (c) incorporate pessimism into learned dynamics models [30, 31]. Since we do not use Decision Transformers to explicitly learn the dynamics model, we primarily compare against model-free algorithms; adding a dynamics model tends to improve the performance of model-free algorithms. Another line of work explores learning wide behavior distribution from an offline dataset by learning a task-agnostic set of skills, either with likelihood-based approaches [32, 33, 34, 35] or by maximizing mutual information [36, 37, 38]. Our work is similar to the likelihood-based approaches, which do not use iterative Bellman updates – although we use a simpler sequence modeling objective instead of a variational method, and use rewards for conditional generation of behaviors.

**Supervised learning in reinforcement learning settings.** Some prior methods for reinforcement learning bear more resemblance to static supervised learning, such as Q-learning [39, 40], which still uses iterative backups, or likelihood-based methods such as behavior cloning, which do not (discussed in previous section). Recent work [8, 9, 10] studies "upside-down" reinforcement learning (UDRL), which are similar to our method in seeking to model behaviors with a supervised loss conditioned on the target return. A key difference in our work is the shift of motivation to sequence modeling rather than supervised learning: while the practical methods differ primarily in the context length and architecture, sequence modeling enables behavior modeling even without access to the reward, in a similar style to language [12] or images [41], and is known to scale well [2]. The method proposed by Kumar et al. [9] is most similar to our method with $K = 1$, which we find sequence modeling/long contexts to outperform (see supplementary material). Ghosh et al. [42] extends prior UDRL methods to use state goal conditioning, rather than rewards, and Paster et al. [43] further use an LSTM with state goal conditioning for goal-conditoned online RL settings. Concurrent to our work, Janner et al. [44] propose Trajectory Transformer, which is similar to Decision Transformer but additionally uses state and return prediction, as well as discretization, which incorporates model-based components.

We believe that their experiments, in addition to our results, highlight the potential for sequence modeling to be a generally applicable idea for reinforcement learning.

**Credit assignment.** Many works have studied better credit assignment via state-association, learning an architecture which decomposes the reward function such that certain "important" states comprise most of the credit [45, 46, 15]. They use the learned reward function to change the reward of an actor-critic algorithm to help propagate signal over long horizons. In particular, similar to our long-term setting, some works have specifically shown such state-associative architectures can perform better in delayed reward settings [47, 7, 48, 26]. In contrast, we allow these properties to naturally emerge in a transformer architecture, without having to explicitly learn a reward function or a critic.

**Conditional language generation.** Various works have studied guided generation for images [49] and language [50, 51]. Several works [52, 53, 54, 55, 56, 57] have explored training or fine-tuning of models for controllable text generation. Class-conditional language models can also be used to learn discriminators to guide generation [58, 50, 59, 60]. However, these approaches mostly assume constant "classes", while in reinforcement learning the reward signal is time-varying. Furthermore, it is more natural to prompt the model desired target return and continuously decrease it by the observed rewards over time, since the transformer model and environment jointly generate the trajectory.

**Attention and transformer models.** Transformers [1] have been applied successfully to many tasks in natural language processing [61, 12] and computer vision [62, 63]. However, transformers are relatively unstudied in RL, mostly due to differing nature of the problem, such as higher variance in training. Zambaldi et al. [5] showed that augmenting transformers with relational reasoning improve performance in combinatorial environments and Ritter et al. [64] showed iterative self-attention allowed for RL agents to better utilize episodic memories. Parisotto et al. [4] discussed design decisions for more stable training of transformers in the high-variance RL setting. Unlike our work, these still use actor-critic algorithms for optimization, focusing on novelty in architecture. Additionally, in imitation learning, some works have studied transformers as a replacement for LSTMs: Dasari and Gupta [65] study one-shot imitation learning, and Abramson et al. [66] combine language and image modalities for text-conditioned behavior generation.

# 7 Conclusion

We proposed Decision Transformer, seeking to unify ideas in language modeling and RL. On standard offline RL benchmarks, we showed DT can match or outperform strong algorithms designed explicitly for offline RL with minimal modifications from standard language modeling architectures.

**Societal impact.** For real-world applications, it is important to understand the types of errors transformers make in MDP settings and possible negative consequences. It will also be important to consider the datasets we train on, which can potentially add destructive biases, particularly as we consider studying augmenting RL agents with more data which may come from questionable sources.

**Limitations.** We introduced our paradigm shift and showed its potential in our experiments, but there is significant room for more research in this direction. The current architecture requires considerations of context length and return-to-go hyperparameters, and we show results on standard RL benchmarks; future work could improve the architecture and demonstrate results in more complex environments and tasks. We used a simple supervised loss that was effective in our experiments, but applications to large-scale datasets could benefit from self-supervised pretraining tasks. In addition, one could consider more sophisticated embeddings for returns, states, and actions. While we do not directly evaluate scaling and generalization, we utilize a method known to scale generalize well in domains such as language and vision, and we are excited about larger RL systems built upon our framework.

# 8 Acknowledgements

This research was supported by Berkeley Deep Drive, Open Philanthropy, and the National Science Foundation under NSF:NRI #2024675. Part of this work was completed when Aravind Rajeswaran was a PhD student at the University of Washington, where he was supported by the J.P. Morgan PhD Fellowship in AI (2020-21). We also thank Luke Metz, Daniel Freeman, and anonymous reviewers for valuable feedback and discussions, as well as Justin Fu for assistance in setting up D4RL benchmarks, and Aviral Kumar for assistance with the CQL baselines and hyperparameters.

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
