# A  Experimental Details

Code for experiments can be found at: `https://github.com/kzl/decision-transformer`.

## A.1  Atari

We build our Decision Transformer implementation for Atari games off of minGPT (`https://github.com/karpathy/minGPT`), a publicly available re-implementation of GPT. We use most of the default hyperparameters from their character-level GPT example (`https://github.com/karpathy/minGPT/blob/master/play_char.ipynb`). We reduce the batch size (except in Pong), block size, number of layers, attention heads, and embedding dimension for faster training. For processing the observations, we use the DQN encoder from Mnih et al. [21] with an additional linear layer to project to the embedding dimension.

For return-to-go conditioning, we use either $1\times$ or $5\times$ the maximum return in the dataset, but more possibilities exist for principled return-to-go conditioning. In Atari experiments, we use Tanh instead of LayerNorm (as described in Section 3) after embedding each modality, but did this does not make a significant difference in performance. We search over $K \in \{1, 10, 30, 50\}$ (see Appendix C for analysis on $K$) and $\widehat{R} \in$ `dataset_max` $\times \{1, 3, 5\}$. The full list of hyperparameters can be found in Table 7.

Table 7: Hyperparameters of DT (and %BC) for Atari experiments.

| Hyperparameter | Value |
|---|---|
| Number of layers | 6 |
| Number of attention heads | 8 |
| Embedding dimension | 128 |
| Batch size | 512 Pong |
| | 128 Breakout, Qbert, Seaquest |
| Context length $K$ | 50 Pong |
| | 30 Breakout, Qbert, Seaquest |
| Return-to-go conditioning | 90 Breakout ($\approx 1\times$ max in dataset) |
| | 2500 Qbert ($\approx 5\times$ max in dataset) |
| | 20 Pong ($\approx 1\times$ max in dataset) |
| | 1450 Seaquest ($\approx 5\times$ max in dataset) |
| Nonlinearity | ReLU, encoder |
| | GeLU, otherwise |
| Encoder channels | $32, 64, 64$ |
| Encoder filter sizes | $8 \times 8, 4 \times 4, 3 \times 3$ |
| Encoder strides | $4, 2, 1$ |
| Max epochs | 5 |
| Dropout | 0.1 |
| Learning rate | $6 * 10^{-4}$ |
| Adam betas | $(0.9, 0.95)$ |
| Grad norm clip | 1.0 |
| Weight decay | 0.1 |
| Learning rate decay | Linear warmup and cosine decay (see code for details) |
| Warmup tokens | $512 * 20$ |
| Final tokens | $2 * 500000 * K$ |

## A.2  OpenAI Gym

### A.2.1  Decision Transformer

Our code is based on the Huggingface Transformers library [67]. Our hyperparameters on all OpenAI Gym tasks are shown below in Table 8. Heuristically, we find using larger models helps to model the distribution of returns, compared to standard RL model sizes (which learn one policy). For reacher we use a smaller context length than the other environments, which we find to be helpful as the environment is goal-conditioned and the episodes are shorter. We choose return targets based on

expert performance for each environment, except for HalfCheetah where we find 50% performance to be better due to the datasets containing lower relative returns to the other environments. Models were trained for $10^5$ gradient steps using the AdamW optimizer [68] following PyTorch defaults. We search over $K \in \{5, 20, 100\}$ (which all yielded similar results), and did not search over $\widehat{R}$ (we conditioned on expert performance for all tasks, except in Cheetah where the dataset did not contain expert data).

Table 8: Hyperparameters of Decision Transformer for OpenAI Gym experiments.

| Hyperparameter | Value |
|---|---|
| Number of layers | 3 |
| Number of attention heads | 1 |
| Embedding dimension | 128 |
| Nonlinearity function | ReLU |
| Batch size | 64 |
| Context length $K$ | 20 HalfCheetah, Hopper, Walker |
| | 5 Reacher |
| Return-to-go conditioning | 6000 HalfCheetah |
| | 3600 Hopper |
| | 5000 Walker |
| | 50 Reacher |
| Dropout | 0.1 |
| Learning rate | $10^{-4}$ |
| Grad norm clip | 0.25 |
| Weight decay | $10^{-4}$ |
| Learning rate decay | Linear warmup for first $10^5$ training steps |

### A.2.2 Behavior Cloning

As briefly mentioned in Section 4.2, we found previously reported behavior cloning baselines to be weak, and so run them ourselves using a similar setup as Decision Transformer. We tried using a transformer architecture, but found using an MLP (as in previous work) to be stronger. We train for $2.5 \times 10^4$ gradient steps; training more did not improve performance. Other hyperparameters are shown in Table 9. The percentile behavior cloning experiments use the same hyperparameters.

Table 9: Hyperparameters of Behavior Cloning for OpenAI Gym experiments.

| Hyperparameter | Value |
|---|---|
| Number of layers | 3 |
| Embedding dimension | 256 |
| Nonlinearity function | ReLU |
| Batch size | 64 |
| Dropout | 0.1 |
| Learning rate | $10^{-4}$ |
| Weight decay | $10^{-4}$ |
| Learning rate decay | Linear warmup for first $10^5$ training steps |

### A.3 Graph Shortest Path

We give details of the illustrative example discussed in the introduction. The task is to find the shortest path on a fixed directed graph, which can be formulated as an MDP where reward is 0 when the agent is at the goal node and $-1$ otherwise. The observation is the integer index of the graph node the agent is in. The action is the integer index of the graph node to move to next. The transition dynamics transport the agent to the action's node index if there is an edge in the graph, while the agent remains at the past node otherwise. The returns-to-go in this problem correspond to negative path lengths and maximizing them corresponds to generating shortest paths.

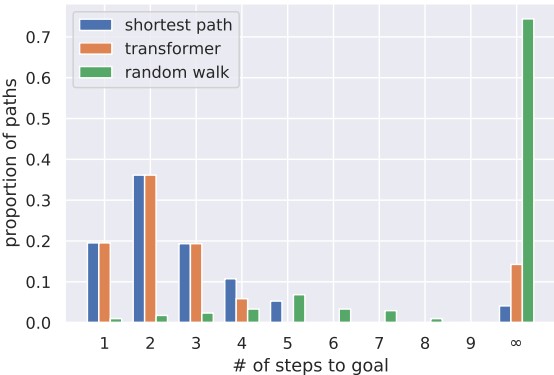

Figure 6: Histogram of steps to reach the goal node for random walks on the graph, shortest possible paths to the goal, and attempted shortest paths generated by the transformer model. $\infty$ indicates the goal was not reached during the trajectory.

In this environment, we use the GPT model as described in Section 3 to generate both actions and return-to-go tokens. This makes it possible for the model it generate its own (realizable) returns-to-go $\hat{R}$. Since we require a return prompt to generate actions and we do assume knowledge of the optimal path length upfront, we use a simple prior over returns that favors shorter paths: $P_{\text{prior}}(\hat{R} = k) \propto T + 1 - k$, where $T$ is the maximum trajectory length. Then, it is combined with the return probabilities generated by the GPT model: $P(\hat{R}_t|s_{0:t}, a_{0:t-1}, \hat{R}_{0:t-1}) = P_{\text{GPT}}(\hat{R}_t|s_{0:t}, a_{0:t-1}, \hat{R}_{0:t-1}) \times P_{\text{prior}}(\hat{R}_t)^{10}$. Note that the prior and return-to-go predictions are entirely computable by the model, and thus avoids the need for any external or oracle information like the optimal path length. Adjustment of generation by a prior has also been used for similar purposes in controllable text generation in prior work [60].

We train on a dataset of $1,000$ graph random walk trajectories of $T = 10$ steps each with a random graph of 20 nodes and edge sparsity coefficient of $0.1$. We report the results in Figure 6, where we find that transformer model is able to significantly improve upon the number of steps required to reach the goal, closely matching performance of optimal paths.

There are two reasons for the favorable performance on this task. In one case, the training dataset of random walk trajectories may contain a segment that directly corresponds to the desired shortest path, in which case it will be generated by the model. In the second case, generated paths are entirely original and are not subsets of trajectories in the training dataset - they are generated from stitching sub-optimal segments. We find this case accounts for $15.8\%$ of generated paths in the experiment.

While this is a simple example and uses a prior on generation that we do not use in other experiments for simplicity, it illustrates how hindsight return information can be used with generation priors to avoid the need for explicit dynamic programming.

# B  Atari Task Scores

Table 10 shows the normalized scores used for normalization used in Hafner et al. [22]. Tables 11 and 12 show the raw scores corresponding to Tables 1 and 4, respectively. For %BC scores, we use the same hyperparameters as Decision Transformer for fair comparison. For REM and QR-DQN, there is a slight discrepancy between Agarwal et al. [16] and Kumar et al. [17]; we report raw data provided to us by REM authors.

| Game | Random | Gamer |
|---|---|---|
| Breakout | 2 | 30 |
| Qbert | 164 | 13455 |
| Pong | −21 | 15 |
| Seaquest | 68 | 42055 |

Table 10: Atari baseline scores used for normalization.

| Game | DT (Ours) | CQL | QR-DQN | REM | BC |
|---|---|---|---|---|---|
| Breakout | **76.9 ± 27.3** | 61.1 | 6.8 | 4.5 | 40.9 ± 17.3 |
| Qbert | 2215.8 ± 1523.7 | **14012.0** | 156.0 | 160.1 | 2464.1 ± 1948.2 |
| Pong | 17.1 ± 2.9 | **19.3** | −14.5 | −20.8 | 9.7 ± 7.2 |
| Seaquest | **1129.3 ± 189.0** | 779.4 | 250.1 | 370.5 | 968.6 ± 133.8 |

Table 11: Raw scores for the 1% DQN-replay Atari dataset. We report the mean and variance across 3 seeds. Best mean scores are highlighted in bold. Decision Transformer performs comparably to CQL on 3 out of 4 games, and usually outperforms other baselines.

## C  Additional Discussions

### C.1  How does context length affect the performance of Decision Transformer?

Results in Atari games with varying context lengths are shown in Table 13. It is important to use $K > 1$ (i.e., to include past context before the current transition). This is interesting since it is generally considered that the previous state (i.e. $K = 1$) is enough for reinforcement learning algorithms when frame stacking is used, as we do. One hypothesis is that when we are representing a distribution of policies – like with sequence modeling – the context allows the transformer to identify which policy generated the actions, enabling better learning and/or improving the training dynamics. $K = 30$ and $K = 50$ generally work well across games, with the exception of Qbert. If one desires to use the same $K$ across all games, $K = 30$ works well. For this ablation, we used $\widehat{R} = 1150$ for Seaquest and $\widehat{R} = 14000$ for Qbert instead of the values in Table 7 but this does not significantly affect performance.

### C.2  Why does Decision Transformer avoid the need for value pessimism or behavior regularization?

One key difference between Decision Transformer and prior offline RL algorithms is that we do not require policy regularization or conservatism to achieve good performance. Our conjecture is that TD-learning based algorithms learn an approximate value function and improve the policy by optimizing this value function. This act of optimizing a learned function can exacerbate and exploit any inaccuracies in the value function approximation, causing failures in policy improvement. Since Decision Transformer does not require explicit optimization using learned functions as objectives, it avoids the need for regularization or conservatism.

### C.3  How can Decision Transformer benefit online RL regimes?

Offline RL and the ability to model behaviors has the potential to enable sample-efficient online RL for downstream tasks. Works studying the transition from offline to online generally find that likelihood-based approaches, like our sequence modeling objective, are more successful [69, 70]. As a result, although we studied offline RL in this work, we believe Decision Transformer can meaningfully improve online RL methods by serving as a strong model for behavior generation. For instance, Decision Transformer can serve as a powerful "memorization engine" and in conjunction with powerful exploration algorithms like Go-Explore [71], has the potential to simultaneously model and generative a diverse set of behaviors.

| Game | DT (Ours) | 10%BC | 25%BC | 40%BC | 100%BC |
|---|---|---|---|---|---|
| Breakout | **76.9 ± 27.3** | 10.0 ± 2.3 | 22.6 ± 1.8 | 32.3 ± 18.9 | 40.9 ± 17.3 |
| Qbert | 2215.8 ± 1523.7 | 1045 ± 232.0 | 2302.5 ± 1844.1 | 1674.1 ± 776.0 | **2464.1 ± 1948.2** |
| Pong | **17.1 ± 2.9** | −20.3 ± 0.1 | −16.2 ± 1.0 | 5.2 ± 4.8 | 9.7 ± 7.2 |
| Seaquest | **1129.3 ± 189.0** | 521.3 ± 103.0 | 549.3 ± 96.2 | 758 ± 169.1 | 968.6 ± 133.8 |

Table 12: %BC scores for Atari. We report the mean and variance across 3 seeds. Decision Transformer usually outperforms %BC.

| Game | $K = 1$ | $K = 10$ | $K = 30$ | $K = 50$ |
|---|---|---|---|---|
| Breakout | 73.9 ± 10 | 183.2 ± 32.1 | **267.5 ± 97.5** | 196.7 ± 106.0 |
| Qbert | 23.0 ± 10.6 | **38.2 ± 11.6** | 25.1 ± 18.1 | 20.9 ± 24.4 |
| Pong | 2.4 ± 0.2 | 32.2 ± 51.9 | 94.6 ± 10.5 | **106.1 ± 8.1** |
| Seaquest | 0.6 ± 0.2 | 2.0 ± 0.4 | **2.4 ± 0.7** | 1.8 ± 0.5 |

Table 13: Results in Atari games with varying context lengths. Decision Transformer generally performs better when using a longer context length.