# OpenReview forum: "Decision Transformer: Reinforcement Learning via Sequence Modeling"
_NeurIPS.cc/2021/Conference — NeurIPS 2021 Poster_

### Official Review · Reviewer_dbvQ · 2021-06-27

**Rating:** 6
**Confidence:** 4

**Summary:**

This paper uses the Transformer model as an auto-regressive model to predict the actions (decisions) givens the state inputs, past actions, and more importantly, the return-to-go. Training such a policy becomes a supervised-learning problem, similar to behavior cloning. The paper applies this method to offline RL problems and demonstrates that policy trained in this supervised manner achieves competitive results compared to SOTA model-free offline RL algorithms.

**Limitations And Societal Impact:**

The paper did not state clearly the limitations of the proposed method. Possible limitations can be on the type of environments where the method can work, the episode length constraint due to the use of attention module, the sensitivity of the hyperparameters such as the training sequence length $K$, etc.

**Main Review:**

This paper is a combination of existing techniques (Transformer + Upside-down RL) and shows that this combination yields better or comparable performance compared to BC and some SOTA offline RL algorithms.

Strengths:
* The method is clearly explained. The paper proposed a simple method to improve behavior cloning performance by conditioning the policy on the return-to-go. Based on the paper writing, one can easily implement the method.

* I like the idea of conditioning the policy on the return-to-go value. In behavior cloning, typically the policy only depends on the states (sometimes also the previous actions). If the training dataset contains multiple state-action pairs on the same state but with different actions, then the policy learning can be hurt as it cannot capture multi-modality. The paper proposes to condition the policy with the return-to-go value as well. With the return value, now the model can have some capacity to fit multiple state-action pairs in the same state because it is very likely that these state-action pairs will have different return values even though the states are the same. This technique can be useful when there is multiple action supervision in the same state or when there is suboptimal action supervision. Note that as the authors also point out in the related work, there are prior works that use this idea such as the Upside-down RL paper.

* The paper does a thorough comparison with SOTA offline RL methods and provides experiments on several environments.


Weaknesses:

* I cannot get why %BC is not a realistic approach. A better explanation would be helpful. In Decision Transformer (DT), one would need to know the return-to-go value to deploy the policy as it's part of the policy input. Then BC can also use this value to filter out the good trajectories on which BC is performed. So it seems that both methods can have the same assumption. If the authors mean that it requires policy deployment to determine whether to use 10%, 25%, 40%, or 100% of the training data, then one can use a threshold return value to filter out the trajectories. So instead of showing the BC performance with %BC, one can show the BC performance trained with datasets that are filtered with different threshold return values.

* This paper seems to over-emphasize the role of the Transformer. From my understanding, conditioning the policy on the return-to-go value seems to be the key to why it can outperform BC. If the authors think that Transformer is the key, then it is necessary to show this with a baseline comparison on LSTM/GRU policies that have the same policy input as the ones used in the paper.

* I would like to see more analysis on why the method works. While there are many experiments showing the performance of DT, little analysis is actually trying to explain why DT works better than BC. Section 5.1 gives some analysis that BC trained with top $X$% of data has similar performance with DT. This could be that DT learns to do BC only on good trajectories and ignore the rest, or DT is able to fit all the trajectories by conditioning the policy on return-to-go value which can tell how good the action is. Also, Is it because DT has the capacity to better deal with multi-modality via the return-to-go policy input?

* An ablation study on the value of the sequence length $K$ is necessary to understand how important this hyperparameter is and is it very sensitive to each game.

* If it is the case that policy in DT can fit all trajectories because the return-to-go value can help the policy distinguish different action supervision even if the state input is very similar, then I would like to see experiments on binary sparse-reward environments. In these environments, the return values for all the successful episodes will be the same (e.g., $+1$) and the return values for all failed episodes will also be the same (e.g., $0$). How does DT perform in these sparse environments?

* The details of the Key-to-Door environments are missing.


**Time Spent Reviewing:**

6

---

> ### Author Response · Authors · 2021-08-10
> **Response to Reviewer dbvQ**
>
> We sincerely thank you for your helpful feedback and insightful comments. We appreciate that our paper is recognized for several positive aspects: (1) clear explanations, (2) usefulness of return-to-go conditioning, and (3) thorough comparison with SOTA offline RL. We address your comments and questions below:
>
> ---
> **Q1. Why %BC is not a realistic approach**
>
> **A1.** Indeed, one could filter trajectories based on return values to avoid choosing an optimal subset size. However, discarding trajectories based on their return is not a good strategy in general, since segments of suboptimal trajectories might still exhibit desirable behavior. DT can still learn from high-return segments of a trajectory even if the trajectory return overall is not high, whereas %BC would discard these, thereby training on a much smaller dataset and being more prone to overfitting. Additionally, we want to emphasize that %BC is a surprisingly strong baseline that, to the best of our knowledge, we are the first to propose for comparison.
>
> **Q2. If Transformer is key, necessary to show LSTM/GRU**
>
> **A2.** As the title suggests, we propose trajectory modeling for RL, and our main contribution is the use of sequence models for RL. We choose the Transformer because it is a particularly promising and efficient architecture. There have been significant engineering efforts to optimize its performance, and there exist strong frameworks and codebases for us to build upon. We propose a generalizable idea that is not limited to Transformers; indeed, we could retrofit our "RL as sequence modeling" idea with future successors of Transformers, and in that sense we envision our method having broad long-term applicability. That said, we will run comparisons to LSTM/GRU for the camera-ready version, since it is an interesting experiment.
>
> **Q3. Why DT works better than %BC**
>
> **A3.** The primary advantage of DT over %BC is in low-data regimes, since %BC discards potentially useful transitions (see A1 above).
>
> **Q4. Ablation on sequence length K**
>
> **A4.** It is important to use $K > 1$ (i.e., to include past context before the current transition). $K = 30$ and $K = 50$ generally work well across games, with the exception of Qbert.
>
> | Game | $K=1$ | $K=10$ | $K=30$ | $K=50$ |
> | :---: | :---: | :---: | :---: | :---: |
> | Breakout | $73.9 \pm 10.0$ | $183.2 \pm 32.1$ | $\bf{267.5} \pm 97.5$ | $196.7 \pm 106.0$ |
> | Qbert | $23.0 \pm 10.6$ | $\bf{38.2} \pm 11.6$ | $25.1 \pm 18.1$ | $20.9 \pm 24.4$ |
> | Pong | $2.4 \pm 0.2$ | $32.2 \pm 51.9$ | $94.6 \pm 10.5$ | $\bf{106.1} \pm 8.1$|
> | Seaquest | $0.6 \pm 0.2$ | $2.0 \pm 0.4$ | $\bf{2.4} \pm 0.7$ | $1.8 \pm 0.5$ |
>
> **Q5. Experiments on binary sparse-reward environments**
>
> **A5.** Key-to-Door is a binary sparse-reward environment. Section 5.5 also studies sparse delayed rewards (although they are not binary)!
>
> **Q6. Key-to-Door details missing**
>
> **A6.** The most important detail is that for the Key-to-Door environment we use the entire episode length as the context, rather than having a fixed content window as in the other environments. We will add this and other details in the camera-ready version!
>
> **Q7. Limitations not clearly stated**
>
> **A7.** We will add a more detailed discussion of the limitations, including those you mentioned, in the camera-ready version.

---

> > ### Comment · Reviewer_dbvQ · 2021-08-19
> > **Response to the rebuttal**
> >
> > Thanks to the authors for their response. The authors do address some of my concerns.
> >
> > Some of the remaining concerns are:
> > * Regarding Q1: It seems that from the table shown in the paper (e.g., Table 3), %BC gets comparative performance to DT. So even if %BC has to discard some of the trajectories (as the authors mentioned in the rebuttal), the final policy performance does not get worse because of this. So it is not a bad thing that %BC has to discard data. Also, more quantitative data is needed to support that %BC does not generalize well due to over-fitting while DT can generalize better because it uses all the training data without discarding any part of the dataset.
> >
> > Also, in practice, when one uses BC and finds it does not work well, one would typically try to filter out some bad data or make the dataset size bigger, etc. Filtering bad data is a common practice and it's similar to %BC. So there is no need to claim credit for being the first to propose this method.
> >
> > I am happy to raise my rating to be 6.

---

> > > ### Author Response · Authors · 2021-08-23
> > > **Response #2 to Reviewer dbvQ**
> > >
> > > Thanks for your response, and we are glad to have addressed some of your concerns! Your feedback has helped us communicate our ideas better, and we will strive to address your remaining concerns.
> > >
> > > Table 3 shows %BC vs DT on Gym tasks, which have low-dimensional inputs and train on a relatively large dataset for state-based tasks (typically 1 million transitions). As a result, it might be possible for %BC to discard 90% of the data and still train successful policies.
> > >
> > > In contrast, Table 4 shows that discarding data in %BC leads to worse performance in Atari games. For tasks with high-dimensional visual inputs and requiring representation learning like Atari, throwing away sub-optimal data can be bad because, at the very least, it can be useful for representation learning.
> > >
> > > Please let us know if there are additional questions, and we are happy to engage in further discussions!

---

### Official Review · Reviewer_8P8v · 2021-07-15

**Rating:** 6
**Confidence:** 4

**Summary:**

This paper proposes a sequence-model approach for offline RL, which does not need to solve cumbersome dynamic programming subproblems. It utilizes a transformer architecture to effectively model state-action sequence which is conditional to the return-to-go. The proposed method shows competitive or superior performance compared to the previous state-of-the-art in various offline RL datasets.

**Limitations And Societal Impact:**

Limitations
- Noted above

Societal Impact
- N/A

**Main Review:**

Pros
- The paper provides a novel and valuable direction to solve RL problems without TD learning which have been fundamental to conventional RL algorithms despite being hard to optimize.
- Transformers have shown remarkable performance across NLP, vision, and GNN. This work introduces a nice way to integrate this architecture to the RL domain.
- The proposed method performs exceptionally well on long-term credit assignments, which seems plausible considering the origin of the transformer architecture.

Cons
- Based on the comparisons with %BC, it seems the proposed method is just a better (or more sample-efficient) way to perform behavior cloning. Can the authors provide some evidence that DT is more than a BC? For example, can the method discover behaviors not directly included in the dataset?
- The authors argue that %BC is not a realistic approach as it needs online evaluations to choose the optimal subset size. However, this goes the same with DT as it needs to tune its hyperparameters ($K$, return-to-go). The appendix shows that the optimal hyperparameters vary between environments.
- The paper does not list the hyperparameter search procedure (search range, selection criteria) at all, which is especially important for offline RL as online evaluations are (conceptually) impossible.
- Sensitivity to the hyperparameters ($K$, return-to-go conditioning) is not given.
- Std values for %BC results are missing in Table 3.
- Exploration is a crucial part for RL algorithms. However, it is unclear how to expand this work to perform explorations with DT.

Other comments
- Does the method fail when it is given an extremely OOD return-to-go value (e.g., $\times 0$, $\times 5$, $\times 10$, ...) ?
- How does the method perform on other D4RL datasets such as *-random and *-full-replay?
- The algorithm is explained only in high-level phrases. Including a pseudo-code of the training and the inference procedure may help the readers understand the method concretely.

========== Update ==========

The author response addresses most of my concerns, and I think the updated manuscript will be valuable to the machine learning community. I am increasing my rating to 6.

**Time Spent Reviewing:**

5

---

> ### Author Response · Authors · 2021-08-10
> **Response to Reviewer 8P8v**
>
> We sincerely thank you for your helpful feedback and insightful comments. We appreciate that our paper is recognized for several positive aspects: (1) novel and valuable direction, (2) nice way to integrate Transformers in RL, and (3) exceptional ability to perform long-term credit assignment. We address your comments and questions below:
>
> ---
> **Q1. Evidence that DT is more than BC**
>
> **A1.** DT can discover behaviors not directly included in the dataset, in the sense that it can sometimes extrapolate to return-to-go values not present in the dataset (Figure 4) and "stitch" suboptimal segments together to generate near-optimal trajectories (Figure 2). A straightforward BC or even %BC algorithm would not be capable of such feats.
>
> **Q2. DT also needs to tune hyperparameters such as K and return-to-go**
>
> **A2.** For return-to-go, choosing the maximum return in the dataset generally works well, and higher numbers are chosen if extrapolation is desired. In addition, returns are human interpretable and it is relatively natural for humans to specify a desired return. We will update the paper to have return-to-go chosen as a function of the dataset, which would implicitly capture domain-specific scales and ranges of return-to-go. Further, DT also does not require training multiple models on different subsets of the data. We agree the return-to-go conditioning could be further improved, and think this would be interesting future work!
>
> For $K$, both DT and %BC need to tune this since we use the same $K$ as DT in our %BC experiments (as it is stronger than $K = 1$).
>
> **Q3. Missing hyperparameter search procedure**
>
> **A3.** In OpenAI Gym, we search over $K \in \\{5, 20, 100\\}$ (which all yielded similar results), and did not search over $\widehat{R}$ (we conditioned on expert performance for all tasks, except in Cheetah where the dataset did not contain expert data). In Atari, we search over $K \in \\{1, 10, 30, 50\\}$ (see A4 for ablation) and $\widehat{R} \in datasetmax \times \\{1, 3, 5\\}$ (the current submission uses CQL performance, but we redid the search to use $datasetmax$ which is cleaner. Updated results, which differ only slightly, are shown below). We agree that moving towards fully offline evaluation and model selection is an important direction for the field of offline RL. For the current work, we emphasize again that our model selection scheme is similar to the approach in many prior offline RL papers (e.g. MOPO, MOReL, COMBO etc).
>
> | Game | DT (Ours) | CQL |
> | :---: | :---: | :---: |
> | Breakout | $267.5 \pm 97.5$ | $211.1$ |
> | Qbert | $15.4 \pm 11.1$ | $104.2$ |
> | Pong | $106.1 \pm 8.1$ | $111.9$ |
> | Seaquest | $2.5 \pm 0.4$ | $1.7$ |
>
> **Q4. Sensitivity to the hyperparameters (K, return-to-go) not given**
>
> **A4.** Sensitivity to return-to-go is given in Figure 4. Sensitivity to $K$ is given in the following table. $K = 30$ and $K = 50$ generally work well across games, with the exception of Qbert. In Gym tasks, results were similar for $K \in \\{5, 20, 100\\}$.
>
> | Game | $K=1$ | $K=10$ | $K=30$ | $K=50$ |
> | :---: | :---: | :---: | :---: | :---: |
> | Breakout | $73.9 \pm 10.0$ | $183.2 \pm 32.1$ | $\bf{267.5} \pm 97.5$ | $196.7 \pm 106.0$ |
> | Qbert | $23.0 \pm 10.6$ | $\bf{38.2} \pm 11.6$ | $25.1 \pm 18.1$ | $20.9 \pm 24.4$ |
> | Pong | $2.4 \pm 0.2$ | $32.2 \pm 51.9$ | $94.6 \pm 10.5$ | $\bf{106.1} \pm 8.1$|
> | Seaquest | $0.6 \pm 0.2$ | $2.0 \pm 0.4$ | $\bf{2.4} \pm 0.7$ | $1.8 \pm 0.5$ |
>
>
> **Q5. Std values for %BC results missing in Table 3**
>
> **A5.** We will fix this!
>
> **Q6. How to perform exploration with DT**
>
> **A6.** We discussed online exploration a bit in Section 5.7. One could model and generate many different behaviors by leveraging the conditional sequence modeling idea. We are really interested in this direction of future work and excited to see extensions of DT to online exploration!
>
> **Q7. Does the method fail when given an extremely OOD return-to-go?**
>
> **A7.** There is some extrapolation ability in Atari games, but in general, DT does not perform well with extremely OOD values (as expected for most deep learning methods).
>
> **Q8. How does the method perform on other D4RL datasets?**
>
> **A8.** We have not tried DT on other D4RL datasets, but will run experiments in *-random and *-full-replay. However, we believe that Medium-Expert, Medium, and Medium-Replay are challenging benchmarks.
>
> **Q9. Including pseudo-code of the training and inference procedure may help**
>
> **A9.** Thanks for the suggestion! We have prepared the pseudocode (https://sites.google.com/view/dt-pseudocode/home) and will add it to the paper.

---

> > ### Comment · Reviewer_8P8v · 2021-08-19
> > **Thank you for the response**
> >
> > I thank the authors for their response. Most of my concerns are addressed. However, I have a few remaining concerns.
> >
> > 1. A1 refers to Figure 4 as evidence that DT sometimes extrapolates to return-to-go values not present in the dataset. Could you explain in more detail which part of the figure shows this extrapolation? The figure seems to show DT's performance is upper-bounded by the best trajectory in the dataset (except for Seaquest, however, the absolute performance seems too low to provide meaningful insights.). I acknowledge understanding the behavior of the learned agents is a problem of RL in general, but I am just curious what you meant by the reference.
> > 2. My primary concern on Q2 was that you mentioned %BC as **not realistic**. The response you provided explains why DT can be a better method than %BC, but still does not address why %BC is unrealistic compared to DT. Both methods require online evaluations for tuning its hyperparameters (e.g., $K$). I worry the current description of %BC in L169~L171 is misleading.

---

> > > ### Author Response · Authors · 2021-08-23
> > > **Response #2 to Reviewer 8P8v**
> > >
> > > Thanks for your response, and we are glad to have addressed most of your concerns! Your feedback has been very helpful in improving our paper, and we will strive to address your remaining concerns below.
> > >
> > > 1. We first wish to clarify that our response was geared towards providing example instances where DT can produce policies better than those in the dataset. Seaquest and Qbert demonstrate this capability, but we agree that absolute performance is low in Seaquest. This capability is more clearly demonstrated in the shortest path graph experiment (see Appendix B), where DT was able to consume data from random walks and produce a policy that is nearly as good as an oracle shortest path algorithm. A straightforward BC or %BC algorithm would not be capable of such feats since they simply learn to mimic the dataset or a subset of it.
> > >
> > > 2. We agree "not realistic" may not be the best characterization of the shortcomings of %BC, and we will rephrase L169-171 to instead highlight the advantages of DT over %BC that we mentioned in A2. Thank you for the suggestion!
> > >
> > > Please let us know if there are additional questions, and we are happy to engage in further discussions!

---

> > > > ### Comment · Reviewer_8P8v · 2021-08-27
> > > > **Final response**
> > > >
> > > > Thank you for the response!
> > > >
> > > > The additional response addresses my remaining concerns, and I think the updated manuscript will be valuable to the machine learning community. I am increasing my rating to 6.

---

### Official Review · Reviewer_BnKN · 2021-07-19

**Rating:** 7
**Confidence:** 4

**Summary:**

The paper investigates the power of Transformers in sequential decision-making tasks. The authors suggest to use trajectory modeling in contrast to using Transformers as a model architecture in conventional RL algorithms. The proposed algorithm is called decision transformer and outperforms model-free offline RL algorithms in tasks requiring long-term credit assignment.

**Limitations And Societal Impact:**

The authors discussed partially the limitations of their approach.
I do not see any potential negative societal impact from this work.


**Main Review:**

I enjoyed reading this paper. The paper is beautifully organized and very well written. Decision Transformer is well motivated and positioned.

The only concern that I have which I think is extremely important is the following:

The authors did not compare Decision Transformers to other related works where Transformers are used as a model architecture within an existing RL method. For instance, the authors did in fact mention (very briefly) the differences of their method compared to reference 43 of the paper in the related works section, but there is no experiment comparing the tradeoff between “Transformers as a model architecture in RL” vs “the paradigm shift proposed here”. Could you please comment? The paper can benefit significantly from an additional experiment here.


**Time Spent Reviewing:**

4

---

> ### Author Response · Authors · 2021-08-10
> **Response to Reviewer BnKN**
>
> We sincerely thank you for your helpful feedback and insightful comments. We appreciate that our paper is recognized for being well-organized, well-written, well-motivated, and well-positioned. We address your comments and questions below:
>
> ---
> **Q1. Comparison to "Transformers as a model architecture in RL"**
>
> **A1.** We agree it would be interesting to compare to these methods, and aim to add comparisons to CQL with Transformers. However, we want to emphasize that these methods do not address fundamental issues with conventional RL. They are susceptible to the same downfalls as conventional RL methods, since they still use RL objectives and bootstrapping to propagate returns. These conventional RL algorithms are often specialized for different domains, and in offline settings, require additional techniques to work effectively. In contrast, we hope that our paradigm shift will allow for simpler, stabler and more general RL frameworks.

---

### Official Review · Reviewer_SzBr · 2021-07-23

**Rating:** 9
**Confidence:** 4

**Summary:**

* This paper introduces Decision Transfomer: a transformer based model that treats RL as a simple sequence modelling problem, and thus is able to eschew dynamic programming bootstrapping involved in contemporary RL algorithms.
* The experiments on Atari and gym environments show that by using an appropriate return as “prompt”, DTs can model the desired behavior, including perform comparably with state-of-the-art offline RL algorithms.


**Main Review:**

This paper proposes a really compelling idea: Can we treat Reinforcement Learning as just another sequence modelling problem? Doing so allows us to re-use all the machinery already developed for modelling sequences in NLP (such as transformers), and removes the need to carefully design RL algorithms.

The core idea is to let transformers model trajectories in an environment where each trajectory is a sequence of returns, states and actions. Interestingly, only predicting actions is found to be necessary (as opposed to predicting other tokens of states and returns as well). Modelling the full trajectory could be a nice direction for future work: something like predicting discretized embeddings of states using a VQ-VAE could work well.  Either way, the idea is indeed simple in hindsight but quite effective: a hallmark of influential papers.

The writing is exceptionally clear throughout, and the authors have put in a good amount work to make the exposition as clear as possible. Figure 1 succinctly describes the idea, and Figure 2 along with the illustrative example does a great job of walking the reader through how the idea would work in a toy setting.

The experiments on Atari are on only 4 games, it would be great if future versions of the paper can expand on this. Some aspects of the experiments do seem problematic:

* **Per-game hyper-parameter tuning**: the context length and desired return ratio are tuned individually for each environment. On Atari, this is not kosher, and the norm is to use a single set of hyper-parameters for all games. The goal of-course is not to build specialized agents for each Atari game, but rather a single algorithm that can master all games, and per-game hyperparam tuning goes against this ethos. This makes comparisons to the baselines unfair / uncontrolled, since I don’t think CQL / REM did per-game hyper-param tuning.
* **Problematic model selection**: It’s not mentioned in the paper, but I imagine the way hyperparams were chosen was by doing *online* policy evaluation. This violates the offline learning assumption as assuming access to online evauluation is often not realistic. [1] and [2] point out these issues and recommend evaluation protocols, so I encourage the authors to show results with offline selection as well.

Minor:
* The introduction should mention upside-down RL since its very similar in similar in spirit, instead of delegating it as another entry in the related work. DTs are a natural evolution of the initial idea proposed in upside-down RL, so it deserves a more central discussion.
* line 109 should say section 5.4 instead of section 5.3

The analysis section is quite thorough and well done, and answers a lot of questions the reader might have about DTs. The sections on modelling diverse returns and sparse rewards are quite informative.

Overall, I think this paper is great fresh new perspective on RL and will undoubtedly be influential. There are some problems with the experimental evaluation in the current version, but nothing that can’t be fixed. I am therefore evaluating assuming these issues can be ironed out.

[1] Gulcehre, C., Wang, Z., Novikov, A., Le Paine, T., Gomez Colmenarejo, S., Zolna, K., ... & de Freitas, N. (2020). Rl unplugged: Benchmarks for offline reinforcement learning. NeurIPS 2020

[2] Paine, Tom Le, Cosmin Paduraru, Andrea Michi, Caglar Gulcehre, Konrad Zolna, Alexander Novikov, Ziyu Wang, and Nando de Freitas. "Hyperparameter selection for offline reinforcement learning." arXiv preprint arXiv:2007.09055 (2020).

**Time Spent Reviewing:**

6

---

> ### Author Response · Authors · 2021-08-10
> **Response to Reviewer SzBr**
>
> We sincerely thank you for your helpful feedback and insightful comments. We appreciate that our paper is recognized for several positive aspects: (1) simplicity yet effectiveness, (2) clear writing, (3) thorough and informative analysis, and (4) potential to be influential. We address your comments and questions below:
>
> ---
> **Q1. Per-game hyperparameter tuning**
>
> **A1.** For context length, the following table shows DT and CQL normalized scores using context length $K = 30$ for each game. We previously used $K = 50$ only for Pong for slightly better results, but performance is still strong for $K = 30$.
>
> | Game | DT (Ours) | CQL |
> | :---: | :---: | :---: |
> | Breakout | $267.5 \pm 97.5$ | $211.1$ |
> | Qbert | $25.1 \pm 18.1$ | $104.2$ |
> | Pong | $94.6 \pm 10.5$ | $111.9$ |
> | Pong ($K=50$) | $106.1 \pm 8.1$ | -- |
> | Seaquest | $2.4 \pm 0.7$ | $1.7$ |
>
> For return-to-go, while raw rewards vary greatly across games, we find that choosing the desired return to be the maximum return in the offline dataset generally works well. Higher values can also be chosen if extrapolating behavior is desired from the agent.
> We believe return-to-go conditioning can potentially be improved even further, and would make for exciting future work!
>
> In addition, there were several hyperparameters we did not tune, such as learning rate, number of self-attention layers, embedding dimension, etc., so there could certainly exist a set of better "universal" hyperparameters.
>
> **Q2. Problematic model selection**
>
> **A2.** We agree that moving towards fully offline evaluation and model selection is an important direction for the field of offline RL. However, we emphasize that this is still very much an open research question, and equally applicable to almost all prior work in offline RL. We hope that the offline RL community as a whole moves towards fully offline approaches. While this would make for great future work, it is unfortunately outside the scope of the current submission. We emphasize again that our model selection scheme is similar to the approach in many prior offline RL papers (e.g. MOPO, MOReL, COMBO etc).
>
> **Q3. Upside-down RL deserves a more central discussion**
>
> **A3.** Thanks for the suggestion! We will add more discussions of upside-down RL in the introduction.

---

> > ### Comment · Reviewer_SzBr · 2021-08-27
> > **Response to rebuttal**
> >
> > Thanks for the additional experiments on hyper-parameter tuning, they are quite informative. For offline model selection, you could refer to RLUnplugged and use their naive baseline (might be too late for this paper but it would strengthen the paper if you could include in the camera ready).

---

### Decision · Program_Chairs · 2021-09-27

**Decision:**

Accept (Poster)

**Comment:**

After an involved discussion on several topics regarding experimental practices and relation to prior art, the reviewers have settled on a range of positive scores. The cited strengths of the paper include the potential impact of introducing modern sequence modeling tools for RL problems, both the offline-RL domain in the paper's experiments and potentially more areas in the future. I personally found several areas where further technical detail and discussion would be desireable in the final version:
A) Expand your analysis of the method's limitations. Note the requests of reviewer dbvQ, but additionally, consider the balance and fairness in your text between identifying the "deadly triad" of deep TD compared to your speculations on the peformance of transformers for the task. The empirical comparison shows DT is strong but not overwhelming and therefore we may find several of the same, or new, problems specific to modeling the relations of reward/return/state/actions over time are present. Help the reader to understand the limits you expect for your method and the community to know when to use this and where to push new methods next.
B) While the training procedure is well documented, some reviewers mentioned the lack of detail about using your model at test/inference time. Even in offline RL benchmarks, one must execute evaluation rollouts where novel states can be encountered and the future return is not available. I'd like to see a full description of the algorithm to perform these rollouts with your model (perhaps in picture form analagous to Fig 1).